# Anserine/Carnosine-Rich Extract from Thai Native Chicken Suppresses Melanogenesis via Activation of ERK Signaling Pathway

**DOI:** 10.3390/molecules27217440

**Published:** 2022-11-02

**Authors:** Karuntarat Teeravirote, Khaetthareeya Sutthanut, Unchalee Thonsri, Panupong Mahalapbutr, Wunchana Seubwai, Sukanya Luang, Patcharaporn Tippayawat, Sakawrat Kanthawong, Chonlatip Pipattanaboon, Monchai Duangjinda, Vibuntita Chankitisakul, Atit Silsirivanit

**Affiliations:** 1Department of Biochemistry, Center for Translational Medicine, Faculty of Medicine, Khon Kaen University, Khon Kaen 40002, Thailand; 2Division of Pharmaceutical Chemistry, Faculty of Pharmaceutical Sciences, Khon Kaen University, Khon Kaen 40002, Thailand; 3Faculty of Medicine, Bangkokthonburi University, Bangkok 10170, Thailand; 4Department of Forensic Medicine, Faculty of Medicine, Khon Kaen University, Khon Kaen 40002, Thailand; 5Department of Medical Technology, Faculty of Associated Medical Sciences, Khon Kaen University, Khon Kaen 40002, Thailand; 6Department of Microbiology, Faculty of Medicine, Khon Kaen University, Khon Kaen 40002, Thailand; 7Department of Animal Science, Faculty of Agriculture, Khon Kaen University, Khon Kaen 40002, Thailand

**Keywords:** chicken, antioxidation, melanin, tyrosinase, L-anserine, L-carnosine, extracellular signal-regulated kinase

## Abstract

Skin hyperpigmentation is an aesthetic problem that leads to psychosocial issues. Thus, skin whitening agents from agro- and poultry-industrial co-products are considered high economic value ingredients of interest for sustainable application. Therefore, this study aimed to determine the cosmeceutical potential of anserine/carnosine-rich chicken extract (ACCE) from the Thai native chicken Pradu Hang Dam Mor Kor 55 (PD) meat. The chemical composition was identified and quantified using the HPLC-UV method. Then, the antioxidation potential of the extract was compared to that of L-anserine and L-carnosine, using 1,1-diphenyl-2-picrylhydrazyl assay and shikonin-induced production of reactive oxygen species in CCD-986Sk cell models, and the anti-melanogenesis effect in the MNT-1 melanoma cell line model was investigated. Furthermore, related mechanisms were identified using colorimetric tyrosinase assay and the Western blot technique. The ACCE was composed of L-anserine and L-carnosine as two major constituents. In a dose-dependent manner, ACCE, L-anserine, and L-carnosine manifested significant antioxidation potential and significant reduction of melanin production. Activation of the extracellular signal-regulated kinase (ERK) signaling pathway and inhibition of tyrosinase activity of ACCE were demonstrated as the mechanisms of the anti-melanogenesis effect. In conclusion, ACCE has been revealed as a potential cosmeceutical agent due to its antioxidation and anti-melanogenic activity in association with L-anserine and L-carnosine composition and biomolecular regulating ability. Therefore, further studies and development should be considered to support the utilization of anserine/carnosine-rich chicken extract in the cosmetic industry for economic value creation and sustainability.

## 1. Introduction

Continuing growth of demands and market values of skin-whitening products has been evidenced, driven by many factors, including the crisis of an aging society, beauty and health awareness among consumers, fashion trends, and environmental pollution. Additionally, the rising consumer awareness regarding the harmful effects of conventional, synthetic chemical whitening agents has created new growth opportunities for organically and naturally derived raw materials and products in the market [1]. This has provided economic value creation from resources containing bioactive agents, including the skin-whitening agents from agro- and poultry-industrial products or co-products, which can serve the market demands and deliver customer satisfaction. Although there are many available chemical tyrosinase inhibitors (whitening agents) in the cosmetic industry, these are used with caution due to risks associated with a range of undesirable side effects including irritation, permanent abnormalities of skin pigmentation, and carcinogenesis [2]. Therefore, research efforts have been ongoing in an effort to discover natural-source alternatives with high safety that are well-characterized, offer multiple bioactivities, and are eco-friendly. Furthermore, poultry-industrial products and co-products are interesting sources of bioactive molecules, such as peptides, hormones, etc. [3,4].

Skin hyperpigmentation is a significant cosmetic concern that possibly affects social life due to an overproduction of melanin pigment. Several factors are known to increase the melanin production in melanocytes, such as ultraviolet (UV) radiation, DNA damage, alpha-melanocyte-stimulating hormone, reactive oxygen species (ROS), and oxidative stress [5,6]. Therefore, multiple pathways related to melanogenesis have been identified as targets for preventing and treating hyperpigmentation disorders. For this purpose, biomolecules such as intracellular ROS and tyrosinase, the key enzyme responsible for melanin production, melanosome formation and transport, and melanogenic cellular signaling pathways are acknowledged biomolecular targets of skin whitening agents, such as tyrosinase inhibitor, antioxidants, and cellular signaling regulators. Based on this information, several natural and synthetic compounds have been studied and used to suppress melanogenesis [7,8,9,10].

Histidine-containing dipeptides, such as L-anserine (β-alanyl-3-methyl-L-histidine) and L-carnosine (β-alanyl-L-histidine), have an antioxidant effect that could detoxify ROS, reactive carbonyl species (RCS), advanced glycoxidation end products (AGEs), and advanced lipoxidation end products (ALEs) [11,12]. Furthermore, these dipeptides also act as a buffer system to balance intracellular pH [13]. This information indicates the health potential and benefits of peptide-rich sources, such as animal-origin extracts. For instance, a chicken extract has been suggested as an alternative health-promoting and anti-aging agent involving its significant antioxidative ability as a primitive bioactivity [14]. Moreover, the protective effect on telomere shortening in fibroblast medium attenuates aging due to the antioxidant, antiglycation, and anti-crosslinking properties of L-carnosine has potentiated the promising application and fruitful outcomes thereof for cosmeceutical purposes [15].

Chicken is an important source of peptides with various beneficial bioactivities, including antioxidation, anti-pancreatic lipase activity, anti-hypertension, alcohol dehydrogenase stabilization, immune boosting, insulin resistance amelioration, and learning enhancement [4,16,17,18,19,20,21]. Moreover, the bioactive agents derived from poultry industrial products or co-products will provide advantages such as unlimited and manageable supplies, a bio-circular and green economy, and industry sustainability. However, there has been limited evidence on the anti-melanogenesis effect and mechanisms of either L-anserine, L-carnosine, or chicken extract. The Thai native chicken Pradu Hang Dam Mor Kor 55 (PD) has been previously shown to be a rich source of L-anserine and L-carnosine [22], which, despite their genuine properties for cosmeceutical benefits, have never been explored. Therefore, scientific supporting data can foster the creation of suitable applications that provide economic value and sustainability.

This study aimed to determine the chemical composition, anti-melanogenesis potential, and related biomolecular targets of Anserine/Carnosine-rich Chicken Extract (ACCE) from the Thai native chicken-PD, focusing on anti-oxidative, anti-tyrosinase, and anti-melanogenesis activity. The obtained data will support the utility and application of animal-origin dipeptides like ACCE in the cosmetic industry with high-economic value and sustainability.

## 2. Results

### 2.1. Anserine/Carnosine-Rich Chicken Extract (ACCE) from Thai Native Chicken-PD Contains a High Content of L-Anserine and L-Carnosine

The ACCE was obtained with a yield of 2.62% *w/w* which was composed of multiple components, including L-anserine and L-carnosine as the two major constituents, with the content of L-carnosine being higher. The ACCE chromatogram at a wavelength of 210 nm indicated more than 10 compounds as the chemical constituents. Amongst these, there were seven major peaks of compounds (peak no. 1–7 in Figure 1A) at various retention times (RTs) and different spectroscopic profiles composited in the ACCE chromatogram. Among them, two major peaks (peak no. 1 and 6) were identified as L-anserine and L-carnosine and confirmed by spike technique (data not shown), while other peaks (peak no. 2–5) were still unknown. Peak #1 at RT 1.50 and UV spectrum with λmax 196 nm (Figure 1A,B) and peak #6 at 5.99 min and UV spectrum with λmax 196 nm (Figure 1A,C) exhibited chromatographic and spectroscopic properties identical to that of the reference standard compounds L-anserine at RT 1.38 min, UV spectrum with λmax 204 nm (Figure 1D,F); and L-carnosine at RT 5.77 min, UV spectrum with λmax 199 nm (Figure 1E,G), respectively. From this, the calibration curves of L-anserine and L-carnosine were successfully developed, which demonstrated different functions of relationship with regression coefficient (R^2^) closed to 1: a linear calibration curve of L-anserine (y = 49056x, R^2^ 0.9988) and a power-function calibration curve of L-carnosine (y = 1689x^0.7191^, R^2^ 0.9942) (Figure 1H). As a result, the L-anserine and L-carnosine contents in the ACCE sample were revealed as 14.43 ± 0.29 µg/mg and 126.02 ± 3.47 µg/mg, respectively.

### 2.2. ACCE, L-Anserine, and L-Carnosine Exerted Antioxidant and Anti-Tyrosinase Activities

The biological activity of ACCE was then analyzed. First, the free radical scavenging activity of ACCE and its active ingredients, L-anserine and L-carnosine, was determined using a 1,1-diphenyl-2-picrylhydrazyl (DPPH) assay. The standard antioxidant, ascorbic acid, was used as the positive control. Compared with the negative control, L-anserine, L-carnosine, and ACCE showed significant free radical scavenging activity in a dose-dependent manner (*p* < 0.05) (Figure 2A). As expected, ascorbic acid, a positive control, could scavenge DPPH with more than 10 times the strength of ACCE and its active ingredients. To confirm the antioxidant activity of ACCE, its ROS scavenging activity on CCD-986Sk human fibroblast cells was measured using chloromethyl-2′,7′-dichlorodihydrofluorescein diacetate (CM-H2DCFDA) staining. Similar to ascorbic acid (the ROS scavenging control, 10 μg/mL), the L-anserine (5 μg/mL), L-carnosine (5 μg/mL), and ACCE (1.0 × 10^4^ μg/mL) could significantly scavenge the ROS produced by 1 μM shikonin stimulation in CCD-986Sk cells (Figure 2B). These results confirm that ACCE exhibited antioxidant activity, potentially relieving ROS- or oxidation-related abnormalities in the cells.

To investigate the effect of ACCE on tyrosinase activity, the in vitro tyrosinase assay confirmed the inhibitory effect of ACCE, L-anserine, and L-carnosine. Similar to arbutin, a reference tyrosinase inhibitor, L-anserine and L-carnosine significantly suppressed cellular tyrosinase activity to 91.60 ± 4.11% and 91.70 ± 5.27% of control, respectively (*p* < 0.05). In addition, ACCE also exhibited anti-tyrosinase activity, demonstrated by the reduction of the cellular tyrosinase activity to 86.93 ± 4.75% of control (*p* < 0.05) (Figure 3), while the histidine, equal concentration with L-anserine, and L-carnosine (1 mM), did not show a significant effect on tyrosinase activity.

### 2.3. ACCE Suppressed Melanin Production via the Activation of the ERK Signaling Pathway in Human Melanoma Cells

To evaluate the anti-melanogenic effect of ACCE and its active ingredients, L-anserine or L-carnosine, the MNT-1 melanoma cell line was treated with 1.0 × 10^4^ and 2.0 × 10^4^ μg/mL of ACCE, 1 and 5 µg/mL of L-anserine and L-carnosine. In addition, arbutin, 2 and 10 µg/mL, was used as an anti-melanogenic positive control. As shown in Figure 4A, arbutin, L-anserine, L-carnosine, and ACCE significantly reduced the melanin production in MNT-1 cells in a dose-dependent manner (*p* < 0.05). Additionally, these compounds did not affect the cell viability of the MNT-1 cells (Figure 4B).

The effect on ERK and Akt signaling pathways was revealed as a molecular target associated with the anti-melanogenesis activity detected in L-anserine, L-carnosine, and ACCE. Compared to the control, L-anserine, L-carnosine, and ACCE drastically activate ERK signaling pathway, as the phosphorylation of ERK at Thr-202 (T202)/ Tyr-204 (Y204) was enhanced at 24 h after treatment (Figure 5). On the other hand, the activation of the Akt signaling pathway via phosphorylation of Akt at Ser-473 was not observed. As shown in Figure 5, the ERK signaling pathway was activated by L-anserine, L-carnosine, and ACCE, contributing to the reduction of melanogenesis. We further examined the effect of ERK suppression on melanin production. Our data showed that, in contrast with ERK activation, the suppression of ERK signaling pathway by MEK inhibitor—PD98059 could significantly enhance the melanin production of MNT-1 cells (Appendix A).

## 3. Discussion

The continuing growth of the market share and demands for natural-origin skin whitening agents and products have potentiated the economic value creation and industrial application of the ACCE. Melanogenesis, a biomolecular-regulating process of melanin production in melanocytes, is an important protective mechanism against UV radiation harm to the skin. At cellular level protection, this attributes to transferring synthesized melanin to the surrounding keratinocytes for scavenging the UV-induced oxidative stress. However, overproduction of melanin has resulted in aesthetic problems from skin disorders and aging, including melasma, freckles, and age spots [10]. As a result, skin whitening agents and products have become high economic value goods, particularly in Asian countries, especially China, India, Korea, and Japan. Many research groups and cosmetic companies have focused on targeting melanogenesis to reduce skin hyperpigmentation and darkness. Several compounds have been discovered and studied to suppress melanogenesis via different mechanisms, such as inhibiting tyrosinase (a rate-limiting enzyme during melanin production), overwhelming melanosome formation, and suppressing melanin transfer [7,8,9,10]. Recently, the extracts from animal products, such as crocodile blood and fish scale, were found to have an anti-melanogenic effect [23,24]. Our ACCE was extracted from chicken meat using an uncomplicated and inexpensive method, which may reflect the lower cost. In addition, comparing Thai native chicken breeds—PD and other Thai native crossbred chickens (Kai-mook and KKU-ONE), the PD native chicken exhibited excellent growth performance, and it is the most popular among consumers because of its texture. Therefore, the number of PD chickens is higher than any other breeds in Thailand. To create a suitable application, economic value, and sustainable utilities, we therefore selected the PD Thai native chicken for our study.

From the result, ACCE has demonstrated beneficiary cosmetic effects due to its ability to scavenge the free radicals and inhibit melanin production in both in vitro and intracellular study models. Moreover, the ability to regulate in multiple biomolecular targets detected in ACCE has interestingly illustrated, L-anserine and L-carnosine―the two of major constituents of ACCE are presumed to play an important role due to their antioxidation activity via free radical scavenging, metal chelating, enhancing the antioxidant enzyme activity, and anti-inflammation [15,25].

Under the HPLC analysis system, the chemical constituents of ACCE were separated and eluted at different retention times and resolutions based on their physicochemical properties (Figure 1A). The chromatogram shows good separating resolution for most of the peaks, complete separating from other peaks, and the peak area is the best choice for quantitative analysis of these good resolution peaks, including L-anserine (peak #1). Unlike the L-anserine, there are partially co-eluted peaks demonstrated in the chromatogram, like peak #6 (of L-carnosine) and peak #7. This has limited the application of peak area (of L-carnosine; #6) for quantitative analysis concerning inaccuracy quantitation caused by the peak overlapping (between peaks #6 and #7). Therefore, the peak heights were used for the L-carnosine quantitation with no interference from the partial co-eluted peak of the compound (peak #7). In addition, due to the asymmetric and skewed shape of the L-carnosine peak with a broadening base when concentration increases (Figure 1E), the mathematical power function (equation y = 1689x^0.7191^) was employed to establish the L-carnosine calibration curve (Figure 1H), delivered an acceptable coefficient of determination (R^2^ = 0.9942). This employed approach has enabled the quantitation of L-carnosine and potentiated the benefits of mathematic models in applications for quantitative chromatographic analysis, particularly for asymmetric peaks and electrical signals [26,27]. However, further development of analytical methods and chemical constituent identification should be considered.

An antioxidant effect of L-anserine, L-carnosine, and ACCE had been reported previously [11,12,22]. The data in our study confirm the ROS scavenging effects of L-anserine and L-carnosine, together with that of ACCE. L-anserine and L-carnosine play an important role in preventing the damages caused by AGEs, ALEs, and oxidative stress [11,12]. ROS production and oxidative stress are closely associated with melanogenesis; suppressing oxidative stress or ROS production contributes to the suppression of melanogenesis [6]. Relatively, from our study result, anti-tyrosinase activity of ACCE, L-anserine, and L-carnosine were illustrated in comparable potencies (Figure 3). This implies that anti-oxidative capacity was shared across the anti-melanogenesis activity of L-carnosine, L-anserine, and other unknown constituents of the ACCE.

In contrast, the anti-melanogenesis effects in MNT-1 melanocytes were differently displayed, with superior activities of L-anserine and L-carnosine over the ACCE. ACCE is a crude extract that contains anserine/carnosine of approximately 10–100 μg/mg protein. However, we speculated that ACCE contains other active ingredients that may benefit on suppression of melanogenesis. From this, the influence of biological circumstance or biodegradable dipeptide properties to hamper the anti-melanogenesis activity of ACCE are assumed and still await further delineation, since scientific data regarding this perspective is currently limited. Thus, further in-depth investigation on their cellular delivery and stability has been necessitated to deliver helpful information for rationale development and application. Remarkably, the important roles of L-anserine and L-carnosine in ACCE bioactivities have been emboldened by the functional study and the effect on ERK signaling pathway in MNT-1 cells (Figure 4 and Figure 5). Moreover, the anti-inflammation and wound healing ability of L-anserine and L-carnosine would fortify the cosmeceutical outcomes of the ACCE [28].

As can be seen in Figure 3, L-anserine, L-carnosine, and ACCE suppressed tyrosinase activity. Similar to L-anserine and L-carnosine, various dipeptides can act as potent melanogenesis and tyrosinase inhibitors [29,30,31]. Although L-anserine and L-carnosine showed relatively small effects on tyrosinase activity (10–15% inhibition), they were found to potently inhibit melanin production in MNT-1 cells (Figure 4). This phenomenon might be explained by our finding that L-anserine and L-carnosine can inhibit melanogenesis via activation of ERK signaling pathway (Figure 5), which may contribute to the suppression of tyrosinase expression in the cells.

Figure 5 demonstrates the effect on the ERK signaling pathway via activation of the ERK signaling pathway to suppress tyrosinase enzyme expression. Our results showed that ACCE and its active ingredients, L-anserine and L-carnosine, could suppress melanogenesis. The mechanisms by which ACCE, L-anserine, and L-carnosine regulate melanogenesis is possibly through the scavenging of ROS to reduce oxidative stress in the cells. Besides oxidative stress, several intracellular signaling pathways can regulate melanogenesis, such as cAMP/PKA, Wnt/β-catenin, MAPK, and PI3K/Akt [6,10]. We have demonstrated that ACCE could activate the ERK signaling pathway, which contributes to suppressing tyrosinase and melanin synthesis in the cells. This evidence suggests that ACCE can suppress melanogenesis via activation of the ERK signaling pathway. We speculated that ACCE activates ERK phosphorylation and may anticipate downregulation of microphthalmia-associated transcription factors (MITF, a critical transcription factor of tyrosinase expression) and tyrosinase. This possibility may need to be explored in the future. Activation of ERK or Akt signaling pathways has been reported to drastically suppress melanogenesis [6,32,33,34,35,36]. Due to Akt signaling not being affected by our ACCE, we therefore performed the ERK inhibition using a MEK inhibitor (PD98059). The results showed that the ERK inhibitor significantly enhances melanogenesis in MNT-1 cells. Our data agree with the previous reports that suppression of ERK or Akt signaling pathways by the specific inhibitors could dramatically enhance melanogenesis [35,37]. Based on the evidence in our study, it is suggested that L-anserine, L-carnosine, and ACCE may suppress melanogenesis by several mechanisms, including anti-oxidative stress, activation of anti-melanogenic signaling (ERK signaling pathway) and suppression of tyrosinase activity. However, besides anserine and carnosine, other metabolites, such as lactate, creatine, alanine, inositol monophosphate (IMP), and inosine, were also identified as the major components of ACCE [22]. We have never analyzed the effect of these metabolites on melanogenesis; hence their anti-melanogenic effect might not be excluded.

In conclusion, these data and information have empowered the ACCE anti-melanogenesis capacity. The overall results have significantly demonstrated ACCE as a potential cosmeceutical active agent due to its anti-oxidative and anti-melanogenesis activity in association with its high content of L-anserine and L-carnosine. Therefore, L-anserine and L-carnosine have been suggested as the potential chemical biomarkers for the quality control of ACCE and products in industrial production.

## 4. Materials and Methods

### 4.1. Cell Line and Cell Culture

MNT-1 (hyperpigmented melanoma cell line) and CCD-986Sk (skin fibroblast cell line) were obtained from the American Type Culture Collection (ATCC, Manassas, VA, USA) and maintained in Dulbecco’s Modified Eagle Medium (DMEM) (Gibco, Brooklyn, NY, USA) supplemented with 10% heat-inactivated fetal bovine serum (FBS) (Gibco, Brooklyn, NY, USA) and 1% antibiotic-antimycotic (Gibco, Brooklyn, NY, USA). The cells were incubated in a 5% CO incubator at 37 °C and subcultured twice a week using 0.05% (*w/v*) trypsin-EDTA (Gibco, Brooklyn, NY, USA). At 80–90% cell confluency, the cells were harvested and processed to the particular assays.

### 4.2. Anserine/Carnosine-Rich Chicken Extract

The anserine/carnosine-rich chicken extract (ACCE) was prepared from breast meat of Thai PD native chickens as previously described [22] with modifications. Briefly, 246 g of chicken breast meat was chopped before homogenizing with 410 mL of distilled water for 5 min. Afterward, the homogenate was centrifuged at 4200× *g* for 15 min at 4 °C. The supernatant was heat-treated at 70–80 °C for 10 min in the water bath. Then the samples were cooled down to room temperature (25–28 °C) before being centrifuged at 4200× *g* twice for 15 min to remove precipitated proteins. The supernatant was filtered through a 0.2 µm membrane and stored at −20 °C until used.

### 4.3. Chemical Composition Using HPLC-UV Analysis

The aqueous solution of the ACCE at a concentration of 0.5 mg/mL and standard compounds (L-anserine and L-carnosine) at various concentrations (0–200 µg/mL) were separately prepared. The analysis was done in an HPLC system of a Surveyor PDA Plus UV detector at wavelengths of 254, 280, and 360 nm, having an RP-C18 column (ZORBAX SB-CN column, particle size 3.5 µm, dimension 4.6 × 150 mm), and a programmatic gradient mobile phase system of methanol and 2% *v/v* formic acid in deionized water mixture at a flow rate of 1 mL/min. The gradient program was comprised of 100% deionized water for 10 min (0.01–10 min), increasing from 0 to 100% methanol within 5 min (10.01–15 min), then increasing the ratio of deionized water to 100% within 5 min (15.01–20 min), respectively. In each run, a 5 µL of the sample (or standard compound) solution was injected into the HPLC system to deliver a chromatogram containing the chemical constituent peak(s) with corresponding retention times (RT), peak area, peak height, and UV spectrum with maximal absorptivity wavelength (λmax). Compared to the peak obtained for L-anserine (or L-carnosine), the reference standard compound, the major peak of the ACCE chromatogram was identified and confirmed by a spike technique; additionally, the content (µg/g extract) of a chemical constituent in the ACCE sample was quantified by using the peak area and peak height. L-anserine content was extrapolated from the calibration curve of the standard L-anserine as the linear plot between average peak areas (y-axis) and corresponding concentrations (x-axis) to deliver a linear equation (y = ax + b) and regression (R^2^). Meanwhile, L-carnosine content was obtained by using peak height to extrapolate from the calibration curve of the standard L-carnosine by plotting between peak heights and corresponding concentrations.

### 4.4. Antioxidative Activity in DPPH Assay

The antioxidative activity of ACCE was determined using a DPPH scavenging assay. Equal volume of L-anserine (0–200 μg/mL, Sigma Aldrich, St. Louis, MO, USA), L-carnosine (0–100 μg/mL, Sigma Aldrich, St. Louis, MO, USA), or ACCE (0–1.0 × 10^4^ μg/mL) was mixed with 0.1 mM DPPH in methanol. The reaction mixture was vortexed thoroughly and incubated at room temperature for 30 min in the dark. The ascorbic acid (0–5 μg/mL, Sigma Aldrich, St. Louis, MO, USA) was used as a standard antioxidant reference. Distilled water was used instead of L-anserine, L-carnosine, ACCE, or ascorbic acid for the control. Equation (1) calculated the percentage (%) of DPPH radical scavenging activity.
% DPPH scavenging activity = [(A0 − A1)/A0] × 100(1)

A0 was the absorbance of the control condition, and A1 was the absorbance of the L-anserine, L-carnosine, or ACCE-treated conditions. The experiment was performed in 3 replicates and repeated at least twice.

### 4.5. ROS Scavenging Assay in CCD-986Sk Human Fibroblast Cells

ROS scavenging activity in CCD-986Sk human fibroblast cells was measured using CM-H2DCFDA (Life Technologies, Invitrogen, Carlsbad, CA, USA) staining, according to the manufacturer’s recommendation. Briefly, the cells were seeded in a 12-well plate, kept overnight, and then treated with 1 μM of ROS-inducer-shikonin (Sigma Aldrich, St. Louis, MO, USA) with/without ascorbic acid (ROS scavenging control, 10 μg/mL) L-anserine (5 μg/mL), L-carnosine (5 μg/mL), ACCE (1.0 × 10^4^ μg/mL) for 24 h. Next, the cells were incubated with 2.5 μM CM-H2DCFDA for 30 min at 37 °C in the serum-free media to measure intracellular ROS level. The fluorescence signal was measured using a flow cytometer, BD Bioscience FACSCanto II (BD Biosciences, Franklin Lakes, NJ, USA). The ROS level was calculated as % of control.

### 4.6. Cellular Tyrosinase Assay

The protocol to measure the effect of ACCE on human tyrosinase activity was modified from the previous study [38]. MNT-1 cells were cultured for 72 h in a 10-cm dish, at 80–90% confluency. The cells were harvested using trypsin–EDTA and washed with ice-cold PBS. The cells were ultrasonicated in 1% Triton X-100, and the lysate was centrifugated at 12,000× *g* for 20 min. The supernatant was collected, and the total protein concentration was determined using the Bradford assay (Bio-Rad Laboratories, Hercules, CA, USA). Approximately 200 µg of protein was mixed with either 2.0 × 10^4^ μg/mL of ACCE, 1 mM (240 μg/mL) L-anserine, or 1 mM (226 μg/mL) L-carnosine in 100 mM potassium phosphate buffer pH 6.8 (total volume 90 µL). Instead of ACCE or inhibitors, PBS was added to the control sample. Then, 10 µL of 50 mM 3,4-Dihydroxy-L-phenylalanine (L-DOPA) (Sigma Aldrich, St. Louis, MO, USA) was added to the mixture and the mixture was incubated at 37 °C for 30 min in the dark. After incubation, the absorbance was measured at 450 nm. The inhibitory effect of each compound on the human tyrosinase activity was compared with the control. Arbutin (272 μg/mL, 1 mM) was used as a tyrosinase inhibitor reference. Equal concentration of histidine (155 μg/mL, 1 mM) was used as a negative control. The experiment was performed in triplicate and repeated at least twice; the data from a representative experiment was presented.

### 4.7. Cell Viability Assay

Cell viability was measured using 3-(4,5-dimethylthiazol-2-yl) 2,5-diphenyltetrazolium bromide (MTT) assay. MNT-1 cells were plated in the 96-well plate (3000 cells/100 µL/well). After overnight incubation, the cells were treated with L-anserine (1 and 5 μg/mL), L-carnosine (1 and 5 μg/mL), ACCE (1.0 × 10^4^ and 2.0 × 10^4^ μg/mL), and ERK inhibitor (PD98059, 1 and 10 μM) for 72 h. After treatment, 10 µL of 5 mg/mL MTT (Invitrogen, Carlsbad, CA, USA) was added and incubated for 3 h. The formazan crystal was solubilized with 110 µL of 0.08 N isopropanol, and the absorbance was measured at 540 nm. Cells treated with PBS instead of ACCE were used as a control. The relative cell number was calculated and presented as % of control. The experiment was performed in 5 replicates and repeated 3 times, and the representative data is shown here.

### 4.8. Melanin Assay

Melanin production was measured using the protocol of the previous study with slight modification [38]. MNT-1 cells were seeded and cultured in 6-well plates for 24 h, then treated with L-anserine, L-carnosine (1 and 5 μg/mL), ACCE (1.0 × 10^4^ and 2.0 × 10^4^ μg/mL) and ERK inhibitor (PD98059, 1 and 10 μM) for five days. Cells treated with phosphate buffer saline (PBS) were used as a control. The cells treated with 2 and 10 μg/mL arbutin (Sigma Aldrich, St. Louis, MO, USA) were used as an anti-melanogenic positive control. After treatment, the cells were detached from the plate with 0.05% trypsin–EDTA and washed with PBS. Melanin contents of approximately 100,000 cells from each condition were solubilized by 100 µL of 2 M NaOH in 10% DMSO for 30 min in the boiling water (100 °C), and the melanin content was measured as the absorbance at 450 nm. The melanin content was calculated as % of control compared with PBS-treated cells.

### 4.9. Western Blot Analysis

MNT-1 cells were cultured in 6-well plates for 24 h then treated with L-anserine, L-carnosine (1 and 5 μg/mL), and ACCE (1.0 × 10^4^ and 2.0 × 10^4^ μg/mL) for 24 or 72 h.; cells treated with PBS were used as a control. The cells were washed with PBS and lysed in cell lysis buffer (1% NP-40, 150 mM NaCl, 50 mM Tris-HCl pH 7.4) containing protease inhibitor and phosphatase inhibitor (Roche, Mannheim, Germany). The cell lysates were run on sodium dodecyl sulphate–polyacrylamide gel electrophoresis (SDS-PAGE) and the proteins were transferred to polyvinylidene fluoride (PVDF) membranes (Millipore; Darmstadt, Germany). Non-specific reactivity was blocked with 5% skim milk in PBS for 1 h. The membranes were then probed with the given concentration of specific primary antibodies; 1:1000 anti-Akt, 1:1000 anti-pAkt (S473), 1:1000 anti-ERK, 1:1000 anti-pERK (T202/Y204) (Cell Signaling Technology; Danvers, MA), 1:2000 anti-tyrosinase (Invitrogen, Carlsbad, CA), and 1:10,000 anti-β-actin (Sigma Aldrich, St. Louis, MO, USA). The signals were visualized using the ECLTM Prime Western Blotting Detection System (AmershamTM, Buckinghamshire, UK) and detected using the ImageQuant LAS 4000 mini-image analyzer and ImageQuantTM TL analysis software (GE Healthcare, Buckinghamshire, UK).

### 4.10. Statistical Analysis

The experimental data was shown as a mean ± standard deviation (SD). The difference between groups was compared using Student’s *t*-test in GraphPad Prism 9.0 software (GraphPad, Inc., La Jolla, CA, USA), *p* < 0.05 was considered statistically significant.

## 5. Conclusions

Anserine/carnosine-rich chicken extract (ACCE) from the Thai native chicken Pradu Hang Dam Mor Kor 55 (PD) meat has been revealed as a potential cosmeceutical agent due to its antioxidation and anti-melanogenic activity in association with L-anserine and L-carnosine composition and biomolecular regulating ability through activation of extracellular signal-regulated kinase (ERK) signaling pathway and inhibition of tyrosinase activity. Therefore, further studies and development should be considered to support the utilization of anserine/carnosine-rich chicken extract in the cosmetic industry to create economic value and sustainability.

## Figures and Tables

**Figure 1 molecules-27-07440-f001:**
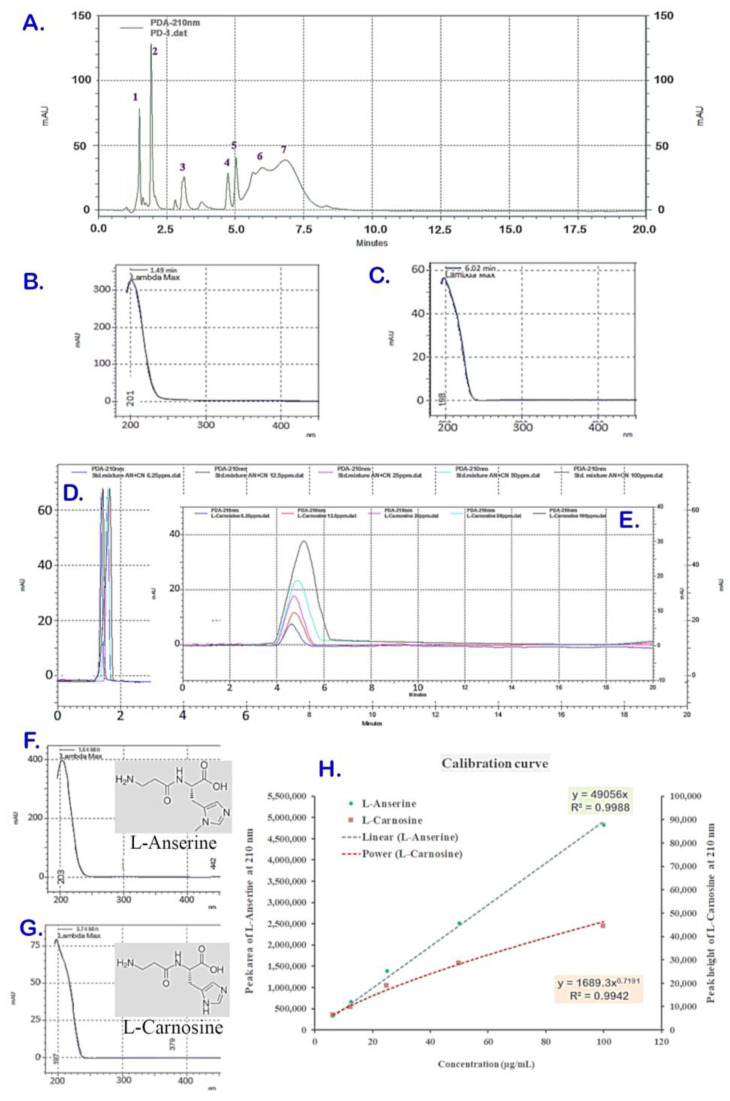
HPLC chromatogram of PD sample containing seven peaks (**A**). Peak #1 was at retention time (RT) 1.50 with a specific UV spectrum (λmax 196 nm) (**B**), and peak #6 was at RT 5.99 min with a particular UV spectrum (λmax 196 nm) (**C**), which were identified as L-anserine and L-carnosine when compared to the standard compounds. The reference standard L-anserine was eluted at RT 1.38 min (**D**) with UV spectrum (λmax 204 nm) (**F**). Meanwhile, L-carnosine was eluted at RT 5.77 min (**E**) with UV spectrum (λmax 199 nm) (**G**). The linear calibration curves of L-anserine plotted between peak areas versus corresponding concentrations and the polynomial calibration curve of L-carnosine plotted between peak heights versus corresponding concentrations (**H**) were established from the chromatograms of various concentrations ((**D**,**E**), respectively).

**Figure 2 molecules-27-07440-f002:**
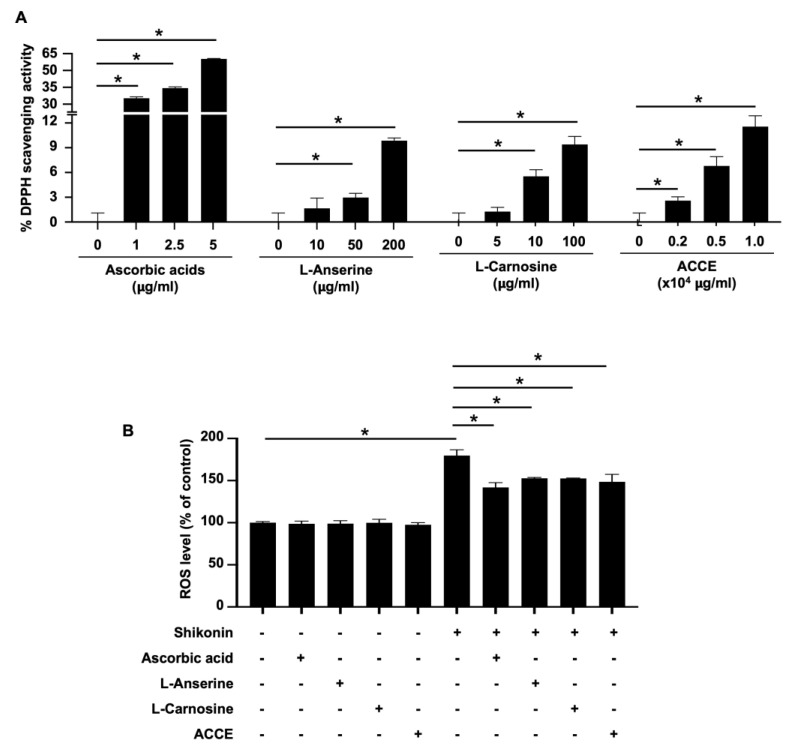
The antioxidant activity of ACCE and its active ingredients, L-anserine and L-carnosine. (**A**) The DPPH scavenging activity of ACCE, L-anserine, and L-carnosine was determined using ascorbic acid as a positive control and methanol diluent as a negative control. (**B**) ROS scavenging activity of ACCE on CCD-986Sk cells was examined using CM-H2DCFDA staining and flow cytometry. Shikonin (was used as the ROS inducer and ascorbic acid was used as a positive control (ROS scavenging agent). Star (*) represents *p* < 0.05 by Student’s *t*-test.

**Figure 3 molecules-27-07440-f003:**
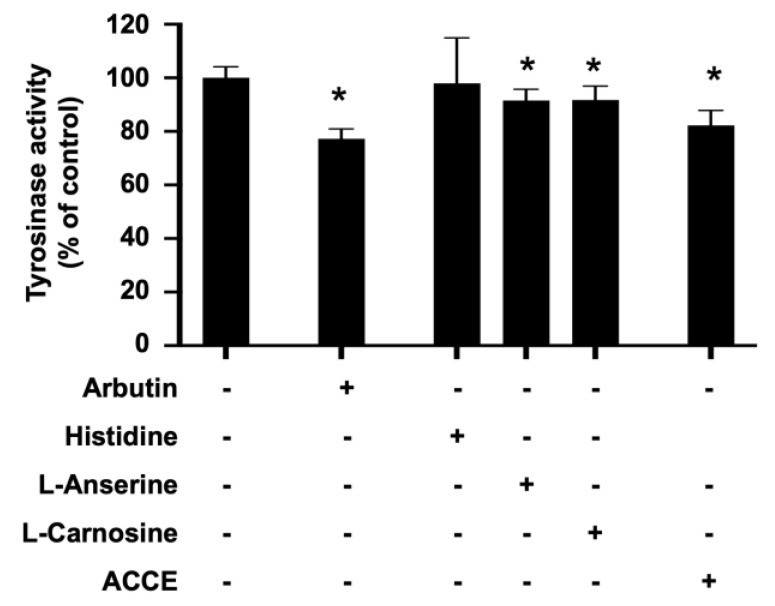
Effect of L-anserine, L-carnosine, and ACCE on cellular tyrosinase activity. Inhibitory effect of ACCE (2.0 × 10^4^ μg/mL), L-anserine (240 μg/mL or 1 mM), and L-carnosine (226 μg/mL or 1 mM) was measured against tyrosinase activity in the extract of MNT-1 cells. Histidine (155 μg/mL or 1 mM) was used as the negative control. Arbutin (272 μg/mL or 1 mM) was used as the positive control. Compared with the non-treated control, the *p* < 0.05 by Student’s *t*-test was defined by the star (*).

**Figure 4 molecules-27-07440-f004:**
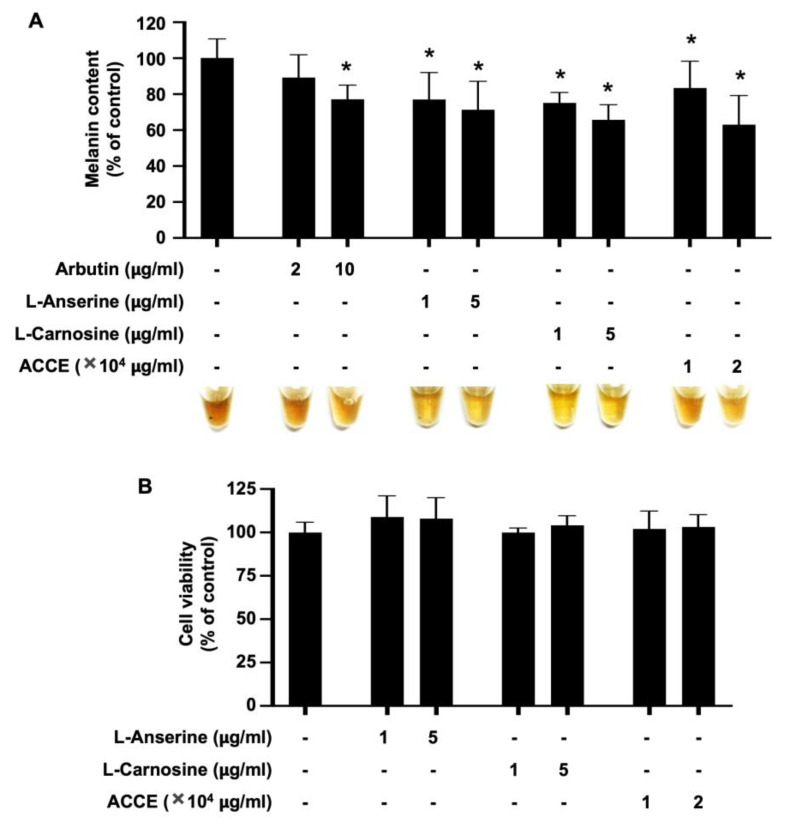
Effect of L-anserine, L-carnosine, and ACCE on melanin production in MNT-1 cells. (**A**) Melanin content in MNT-1 cells was measured at 72 h after treatment with 2 and 10 µg/mL of Arbutin, 1 and 5 µg/mL of L-anserine or L-carnosine, and 1.0 × 10^4^ and 2.0 × 10^4^ μg/mL of ACCE. (**B**) Cell viability of MNT-1 cells was measured at 72 h after treatment with ACCE using MTT assay. Star (*) represents *p* < 0.05 by Student’s *t*-test.

**Figure 5 molecules-27-07440-f005:**
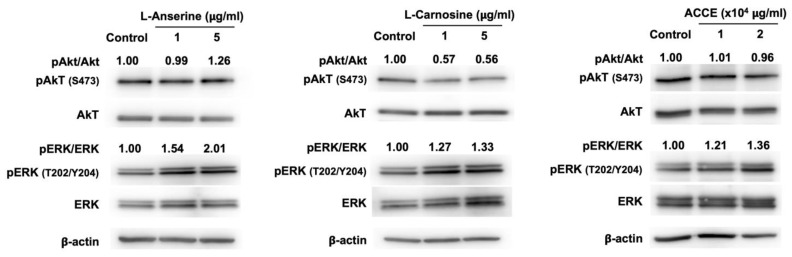
Effect of L-anserine, L-carnosine, and ACCE on melanogenesis-related signaling pathway. Western blot analysis was performed to determine the ratio of phosphorylated and non-phosphorylated Akt and ERK in MNT-1 cells at 24 h.

## Data Availability

Not applicable.

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
