# Peer review of "Anserine/Carnosine-Rich Extract from Thai Native Chicken Suppresses Melanogenesis via Activation of ERK Signaling Pathway"

_molecules, 2022, doi:10.3390/molecules27217440_

Round 1
Reviewer 1 Report
The study from Teeravirote et al aimed to determine the cosmeceutical potential of Anserine/Carnosine-rich Chicken Extract (ACCE) for anti-melanogenesis effect via the activation of the ERK signalling pathway.
Through quantitation and identification of components from ACCE using HLPC, the authors selected two major chemicals from ACCE for the further analysis of anti-oxidative, anti-tyrosinase, and anti-melanogenesis activity. Then, Teeravirote et al showed these two extracts from ACCE, L-Anserine and L-Carnosine, docked and inhibited human tyrosinase activities by molecular docking simulation and in vitro tyrosinase assay, respectively. Finally, the authors revealed the molecular mechanism that ACCE and its extracts suppressed melanin production via the activation of the ERK signalling pathway without cytotoxicity to cells. The experiment design was reasonable, complete and consecutive.
However, several issues need to be fixed and enhanced:
-
there are several questions that remained in Figure 1.
-
the quality of images is poor and higher resolution requires
-
since the peak of No.7 extraction is merged with No. 6, it is questionable for the quantitation of No. 6 extraction.
-
It is difficult to distinguish detailed differences based on the current quality of the image in Fig 1D. Thus, a higher resolution of the figure is needed
-
In line 129, the text of Figure 1C is missing
2. In the abstract, Teeravirote et al mentioned at least two cell lines were used in this study, however, some figures failed to label the specific cell lines used for corresponding analysis.
3. Figure legends are deficient in Figure 6
4. For further validation, authors could compare the biological behaviour differences between ERK inhibition and L-Anserine or L-Carnosine treatment. Moreover, the rescue experiment could be established for the inhibitory efficiency of L-Anserine or L-Carnosine in the cells treated with an AKT activator, such as SC79.
Author Response
Reviewer#1
The study from Teeravirote et al aimed to determine the cosmeceutical potential of Anserine/Carnosine-rich Chicken Extract (ACCE) for anti-melanogenesis effect via the activation of the ERK signalling pathway.
Through quantitation and identification of components from ACCE using HLPC, the authors selected two major chemicals from ACCE for the further analysis of anti-oxidative, anti-tyrosinase, and anti-melanogenesis activity. Then, Teeravirote et al showed these two extracts from ACCE, L-Anserine and L-Carnosine, docked and inhibited human tyrosinase activities by molecular docking simulation and in vitro tyrosinase assay, respectively. Finally, the authors revealed the molecular mechanism that ACCE and its extracts suppressed melanin production via the activation of the ERK signalling pathway without cytotoxicity to cells. The experiment design was reasonable, complete and consecutive.
However, several issues need to be fixed and enhanced:
- there are several questions that remained in Figure 1.
the quality of images is poor and higher resolution requires
Response: Thank you very much. Figure 1 has been improved accordingly, replacement with high resolution pictures.
since the peak of No.7 extraction is merged with No. 6, it is questionable for the quantitation of No. 6 extraction.
Response: In the case of quantitative analysis of L-carnosine (#6), using the peak heights and delivering the power function calibration curve (plotted between peak heights and concentrations) are considered acceptable and suitable, rather than using the peak area which will lead to inaccurate quantitation from the peak overlapping (between peaks #6 and #7), interference from the peak #7.
- The peak height is the most acceptable and suitable variable to be used instead of the peak area : The similar physicochemical properties the L-carnosine (#6) and an unknown (#7) have resulted difficulty to separate the peaks of the L-carnosine (#6) and an unknown (#7)―the peak overlapping (between peaks #6 and #7) has been obtained in the chromatogram which limited the application of peak area (of L-carnosine; #6) for quantitative analysis. However, the more accurate quantitation of the L-carnosine (peak #6) is enabled by using the peak height to extrapolate the concentration from the standard calibration curve with no interference from peak #7 (the partial co-eluted peak).
- Under the HPLC analysis system, the chemical constituents of ACCE were separated and eluted at different retention times and resolutions based on their physicochemical properties. The chromatogram shows good separating resolution for most of the peaks, complete separating from other peaks, and peak area is the best choice for quantitative analysis of these good resolution peaks, including L-anserine (peak #1). Unlike the L-anserine, there are partially co-eluted peaks demonstrated in the chromatogram, like peak#6 (of L-carnosine) and peak#7. This has limited the application of peak area (of L-carnosine; #6) for quantitative analysis concerning inaccuracy quantitation caused by the peak overlapping (between peaks #6 and #7). Therefore, instead of peak areas, the peak heights were used for the L-carnosine quantitation with no interference from the partial co-eluted peak of the compound (peak#7).
It is difficult to distinguish detailed differences based on the current quality of the image in Fig 1D. Thus, a higher resolution of the figure is needed
Response: This has been improved accordingly by the higher resolution pictures.
In line 129, the text of Figure 1C is missing
Response: This has been corrected accordingly.
- In the abstract, Teeravirote et al mentioned at least two cell lines were used in this study, however, some figures failed to label the specific cell lines used for corresponding analysis.
Response: Thank you very much for the constructive encouragement, we have carefully checked and defined the name of cell lines used for each experiment.All figures presented the data using cell lines were now labeled.
Figure legends are deficient in Figure 6
Response: Thank you very much for the constructive encouragement, all figures are now with legends.
For further validation, authors could compare the biological behaviour differences between ERK inhibition and L-Anserine or L-Carnosine treatment. Moreover, the rescue experiment could be established for the inhibitory efficiency of L-Anserine or L-Carnosine in the cells treated with an AKT activator, such as SC79.
Response: Thank you very much for your kind suggestion. Activation of ERK or Akt signaling pathways have been reported to drastically suppress the melanogenesis [6, 30-34]. Due to Akt signaling was not affected by our ACCE, we therefore performed the ERK inhibition using MEK inhibitor (PD98059). The results showed that ERK inhibitor significantly enhances the melanogenesis in MNT-1 cells. Our data agrees with the previous reports that suppression of ERK or Akt signaling pathway by the specific activators could significantly enhance the melanogenesis [33, 35]. We have added this information in the results and discussion.
Results, Page 7, Line 201-206 and the supplementary Fig.:
“As shown in Figure 5, the ERK signaling pathway was activated by L-anserine, L-carnosine, and ACCE, contributing to the reduction of melanogenesis. We further examined the effect of ERK suppression on melanin production. Our data showed that, in contrast with ERK activation, the suppression of ERK signaling pathway by MEK inhibitor-PD98059 could significantly enhance the melanin production of MNT-1 cells (supplementary data).”
Discussion, Page 9, Line 287-293:
Activation of ERK or Akt signaling pathways has been reported to drastically suppress melanogenesis [6, 30-34]. Due to Akt signaling not being affected by our ACCE, we therefore performed the ERK inhibition using a MEK inhibitor (PD98059). The results showed that ERK inhibitor significantly enhances melanogenesis in MNT-1 cells. Our data agree with the previous reports that activation of ERK or Akt signaling pathway by the specific activators could dramatically suppress melanogenesis [33, 35].

Reviewer 2 Report
In this manuscript, Karuntarat Teeravirote et al. investigated the composition and bio-properties of a rich extract from Thai native chicken. It could mean that the authors worked hard to get all these results. However, there are some defects that the authors need to clarify or address before this paper can be published.
1. There are no clear reasons why the authors decided to evaluate the Thai native chicken-Pradu Hang Dam Mor Kor 55 instead of other crossbred chickens. In reference no.22, many Thai native chickens have been previously demonstrated as a rich source of L-Anserine and L-Carnosine.
2. For chemical composition, the HPLC chromatogram of the Anserine/Carnosine-rich Chicken Extract (ACCE) is not clear. The resolution of peak no.6 is very poor. The authors have to develop a more effective technique in order to know the real quantity of L-carnosine and try to avoid a polynomial calibration curve.
3. To confirm the main peptide composition, other methods such as LCMS or NMR should be utilized.
4. For antioxidant activity, the ACCE, L-carnosine, and L-anserine showed weak activity when compared with ascorbic acid by the DPPH assay. In ROS scavenging activity, the authors should add positive control, L-carnosine, and L-anserine to the experiment.
5. For anti-tyrosinase activity, the ACCE had lower activity than arbutin, L-carnosine, and L-anserine by about 10,000 times. The authors should discuss this point.
6. In the discussion, the authors should compare the strength of ACCE activity with that of other studies, especially those from animal product extracts.
7. The authors should report the percent yield of the ACCE.
8. The authors state that content of a chemical constituent in the ACCE sample was quantified by using the peak area and peak height. Normally, when the quantity of chemicals is evaluated, the peak area or peak height will be chosen for calculation. The authors should explain how to calculate the quantity of L-carnosine and L-anserine when using both parameters.
Author Response
Reviewer#2
In this manuscript, Karuntarat Teeravirote et al. investigated the composition and bio-properties of a rich extract from Thai native chicken. It could mean that the authors worked hard to get all these results. However, there are some defects that the authors need to clarify or address before this paper can be published.
- There are no clear reasons why the authors decided to evaluate the Thai native chicken-Pradu Hang Dam Mor Kor 55 instead of other crossbred chickens. In reference no.22, many Thai native chickens have been previously demonstrated as a rich source of L-Anserine and L-Carnosine.
Response: Thank you very much for the constructive comments. Comparing among Thai native chicken breeds (PD and Chee) and Thai native crossbred chickens (Kai-mook and KKU-ONE), the ACCE concentrations were not different as the reviewer’s mentioned. Thus, it supposed that any breeds could be used as ACCE resources. However, PD native chicken exhibited excellent growth performance while CH native chicken was superior in egg production. Besides, PD is the most popular for the consumers because of their texture. Therefore, the number of PD chickens is higher than any other breeds in Thailand. Meanwhile Kai-mook and KKU-ONE are Thai synthetic chicken (TSCs). To create a suitable application, economic value, and sustainable utilities, we therefore selected the PD chicken which is Thai native chicken in our study.
We have added this information in the Discussion, Page 8, Line 232-237 as
“In addition, comparing Thai native chicken breeds-PD and other Thai native crossbred chickens (Kai-mook and KKU-ONE), the PD native chicken exhibited excellent growth performance, and it is the most popular among consumers because of its texture. Therefore, the number of PD chickens is higher than any other breeds in Thailand. To create a suitable application, economic value, and sustainable utilities, we therefore selected the PD Thai native chicken for our study.”
- For chemical composition, the HPLC chromatogram of the Anserine/Carnosine-rich Chicken Extract (ACCE) is not clear. The resolution of peak no.6 is very poor. The authors have to develop a more effective technique in order to know the real quantity of L-carnosine and try to avoid a polynomial calibration curve.
Response: Thank you very much for the constructive suggestion, In the case of quantitative analysis of L-carnosine, using the peak heights are considered acceptable and suitable, which the plot between peak heights and concentrations delivered the power function calibration curve (equation y = 1689x0.7191, R2 0.9942; L124) for the L-carnosine quantitation.
- The peak height is the most acceptable and suitable variable to be used instead of the peak area : The similar physicochemical properties the L-carnosine (#6) and an unknown (#7) have resulted in difficulty to separate the peaks of the L-carnosine (#6) and an unknown (#7)―the peak overlapping (between peaks #6 and #7) has been obtained in the chromatogram which limited the application of peak area (of L-carnosine; #6) for quantitative analysis. However, the more accurate quantitation of the L-carnosine (peak #6) with no interference from the peak #7 (the partial co-eluted peak) is enabled by using the peak height to extrapolate the concentration from the standard calibration curve with no interference from the peak #7 (the partial co-eluted peak).
- The power function calibration curve of L-carnosine is used instead of polynomial function. As the result, the revision of Figure 1H and re-calculation for the L-Carnosine content in ACCE using the power function equation calibration curve was performed.
- To confirm the main peptide composition, other methods such as LCMS or NMR should be utilized.
Response: Thank you very much for your informative suggestion, we agree with the reviewer’s suggestion. This part of research work is in our further plan that we are planning to use other analytical methods to identify the unknown peaks. From the results, L-Anserine and L-Carnosine are the two of major constituents in ACCE extract. Therefore, the phrases in Results, Page 3, Line 107 and Page 8, Line 241-242, were improved as “the two of major constituents in ACCE extract.”
In addition, the composition of PD-1 meat has previously been characterized by NMR in Charoensin et al (Ref. 22). Lactate, anserine, carnosine, creatine, alanine, inositol monophosphate (IMP), and inosine, were identified as the major components. We have added this information in the
Discussion Page 9, Line 296-300, as:
“However, besides anserine and carnosine, other metabolites, such as lactate, creatine, alanine, inositol monophosphate (IMP), and inosine, were also identified as the major components of ACCE [22]. We have never analyzed the effect of these metabolites on melanogenesis; hence their anti-melanogenic effect might not be excluded. ”
- For antioxidant activity, the ACCE, L-carnosine, and L-anserine showed weak activity when compared with ascorbic acid by the DPPH assay. In ROS scavenging activity, the authors should add positive control, L-carnosine, and L-anserine to the experiment.
Response: Thank you very much for your valuable suggestion. We have analyzed the ROS scavenging activity of ascorbic acid (the ROS scavenging control), L-carnosine, and L-anserine as your suggestion and add the data into the manuscript as follows.
Results, Page 5, Line 147-150, as:
“Similar to ascorbic acid (the ROS scavenging control, 10 mg/ml), the L-Anserine (5 mg/ml), L-Carnosine (5 mg/ml), and ACCE (1.0 x 104 mg/ml) could significantly scavenge the ROS produced by 1 mM shikonin stimulation in CCD-986Sk cells (Figure 2B).”
Figure 2B, Page 5 was re-arranged.
- For anti-tyrosinase activity, the ACCE had lower activity than arbutin, L-carnosine, and L-anserine by about 10,000 times. The authors should discuss this point.
Response: Thank you very much for your kind suggestion. ACCE is a crude extract that contains anserine/carnosine of approximately 10-100 mg/mg protein. Therefore, the efficacy of ACCE on anti-tyrosinase and melanogenesis are relatively lower than arbutin, pure L-carnosine, and pure L-anserine. However, we speculated that ACCE contains other active ingredients that may benefit on suppression of melanogenesis. In addition, ACCE is extracted from PD chicken meat by an uncomplicated and inexpensive method, which may reflect the lower cost compared with pure compounds. Moreover, it not only benefits cosmetic sciences but using ACCE may also help the social by broadening the application of PD chicken in many aspects.
We have discussed this point in the Discussion, Page 9, Line 255-264 as
“In contrast, their anti-melanogenesis effects in MNT-1 melanocytes were differently displayed; superior activities of L-Anserine and L-Carnosine over the ACCE. ACCE is a crude extract that contains anserine/carnosine of approximately 10-100 mg/mg protein. However, we speculated that ACCE contains other active ingredients that may benefit on suppression of melanogenesis. From this, the influence of biological circumstance or biodegradable dipeptides properties to hamper the anti-melanogenesis activity of ACCE are assumed and still await further delineation since scientific data regarding this perspective has been currently limited. Thus, further in-depth investigation on their cellular delivery and stability has been necessitated to deliver helpful information for rationale development and application.”
- In the discussion, the authors should compare the strength of ACCE activity with that of other studies, especially those from animal product extracts.
Response: Thank you very much for your valuable suggestion. Recently, the extracts from animal products, such as crocodile white blood cells and fish scale, were found to have the anti-melanogenic effect. However, our ACCE was extracted from chicken meat using an uncomplicated method. Moreover, breeding of PD chicken is easy, therefore, the number of PD chickens is higher than any other breeds in Thailand. Using our ACCE may promote a suitable application, economic value, and sustainable utilities.
We have added this information in the Discussion, Page 8, Line 229-237 as
“Recently, the extracts from animal products, such as crocodile blood and fish scale, were found to have an anti-melanogenic effect [23, 24]. Our ACCE was extracted from chicken meat using an uncomplicated and inexpensive method, which may reflect the lower cost. In addition, comparing Thai native chicken breeds-PD and other Thai native crossbred chickens (Kai-mook and KKU-ONE), the PD native chicken exhibited excellent growth performance, and it is the most popular among consumers because of its texture. Therefore, the number of PD chickens is higher than any other breeds in Thailand. To create a suitable application, economic value, and sustainable utilities, we therefore selected the PD Thai native chicken for our study.”
- The authors should report the percent yield of the ACCE.
Response: %yield is 2.62 w/w (PD-1, chicken breast, 1 kg gives ACCE 26.20 g). This has been added in the Results, Page 3, Line 106.
- The authors state that content of a chemical constituent in the ACCE sample was quantified by using the peak area and peak height. Normally, when the quantity of chemicals is evaluated, the peak area or peak height will be chosen for calculation. The authors should explain how to calculate the quantity of L-carnosine and L-anserine when using both parameters.
Response: Under the HPLC analysis system, the chemical constituents of ACCE were separated and eluted at different retention times and resolutions based on their physicochemical properties. The chromatogram shows good separating resolution for most of the peaks, complete separating from other peaks, and the peak area is the best choice for quantitative analysis of these good resolution peaks, including L-anserine (peak #1). Unlike the L-anserine, there are partially co-eluted peaks demonstrated in the chromatogram, like peak#6 (of L-carnosine) and peak#7. This has limited the application of peak area (of L-carnosine; #6) for quantitative analysis concerning inaccuracy quantitation caused by the peak overlapping (between peaks #6 and #7). Therefore, the peak heights were used for the L-carnosine quantitation with no interference from the partial co-eluted peak of the compound (peak #7).

Round 2
Reviewer 1 Report
The authors have actively made corresponding improvements in response to the questions raised, and the results and discussion sections have been updated. For Figure 1, which explains why peak areas are used for quantification instead of peak areas, Figure 1 D replaces the high-resolution graph; The cell line names in the figure are indicated; According to the suggestion, the authors performed rescue experiments with MEK inhibitors to verify the toxicity and melanogenesis of MNT-1 cell line after the addition of inhibitors
Author Response
Reviewer#1
The authors have actively made corresponding improvements in response to the questions raised, and the results and discussion sections have been updated. For Figure 1, which explains why peak areas are used for quantification instead of peak areas, Figure 1 D replaces the high-resolution graph; The cell line names in the figure are indicated; According to the suggestion, the authors performed rescue experiments with MEK inhibitors to verify the toxicity and melanogenesis of MNT-1 cell line after the addition of inhibitors.
Response: Thank you very much for your kind comments and suggestions.
Reviewer 2 Report
Response number 8 and the authors' references to the HPLC analysis system should be included in the discussion.
Author Response
Reviewer#2
Response number 8 and the authors' references to the HPLC analysis system should be included in the discussion.
Response: Thank you very much for your kind suggestions. This was added as the reviewer’s suggestion, please see in Discussion, Page 8, Line 246-L263 as
“Under the HPLC analysis system, the chemical constituents of ACCE were separated and eluted at different retention times and resolutions based on their physicochemical properties (Figure 1A). The chromatogram shows good separating resolution for most of the peaks, complete separating from other peaks, and the peak area is the best choice for quantitative analysis of these good resolution peaks, including L-anserine (peak #1). Unlike the L-anserine, there are partially co-eluted peaks demonstrated in the chromatogram, like peak#6 (of L-carnosine) and peak#7. This has limited the application of peak area (of L-carnosine; #6) for quantitative analysis concerning inaccuracy quantitation caused by the peak overlapping (between peaks #6 and #7). Therefore, the peak heights were used for the L-carnosine quantitation with no interference from the partial co-eluted peak of the compound (peak #7). In addition, due to the asymmetric and skewed shape of the L-carnosine peak with a broadening base when concentration increases (Figure 1E), the mathematical power function (equation y = 1689x0.7191) was employed to establish the L-carnosine calibration curve (Figure 1H), delivered an acceptable coefficient of determination (R2 = 0.9942). This employed approach has enabled the quantitation of the L-carnosine and potentiated the benefits of mathematic models in applications for quantitative chromatographic analysis, particularly for asymmetric peaks and electrical signals [26,27]. However, further development of analytical methods and chemical constituent identification should be considered.”